# Workflow and Strategies for Recruitment and Retention in Longitudinal 3D Craniofacial Imaging Study

**DOI:** 10.3390/ijerph16224438

**Published:** 2019-11-12

**Authors:** Rafael Denadai, Junior Chun-Yu Tu, Ya-Ru Tsai, Yi-Ning Tsai, Emma Yuh-Jia Hsieh, Betty CJ Pai, Chih-Hao Chen, Alex Kane, Lun-Jou Lo, Pang-Yun Chou

**Affiliations:** 1Department of Plastic and Reconstructive Surgery and Craniofacial Research Center, Chang Gung Memorial Hospital, Chang Gung University, Taoyuan 333, Taiwan; denadai.rafael@hotmail.com (R.D.); jr_tu@hotmail.com (J.C.-Y.T.); chchen5027@gmail.com (C.-H.C.); lunjoulo@cgmh.org.tw (L.-J.L.); 2Image Lab, Craniofacial Research Center, Chang Gung Memorial Hospital, Taoyuan 333, Taiwan; j69278059@gmail.com; 3Wushin Mental Health Foundation, Taipei 104, Taiwan; cass541020@gmail.com; 4Division of Craniofacial Orthodontics, Department of Dentistry, Chang Gung Memorial Hospital, Taoyuan 333, Taiwan; emma.jia@gmail.com (E.Y.-J.H.); pai0072@cgmh.org.tw (B.C.P.); 5Analytical Imaging and Modeling Center, Department of Plastic Surgery, University of Texas Southwestern, Dallas, TX 75390, USA; alex.kane@utsouthwestern.edu

**Keywords:** longitudinal study design, workflow, recruitment, retention, 3D image, normative dataset

## Abstract

Longitudinal epidemiological studies are considered the gold standard for understanding craniofacial morphologic development, but participant recruitment and retention can be challenging. This study describes strategies used to recruit and maintain a high level of participation in a longitudinal study involving annual three-dimensional (3D) craniofacial soft-tissue imaging from healthy Taiwanese Chinese elementary school students aged 6 to 12 years. The key aspects for project delineation, implementation, and the initial three-year practical experiment are portrayed in an integrated multistep workflow: ethics- and grant-related issues; contact, approval, and engagement from partners of the project (school stakeholders and parents); a didactic approach to recruit the students; research staff composition with task design; three station-based data collection days with two educative activities (oral hygiene and psychosocial interaction stations) and one 3D craniofacial imaging activity; and reinforcement tactics to sustain the longitudinal annual participation after the first enrollment. Randomly selected students and teachers answered an experience satisfaction questionnaire (five-point Likert scale ranging from one to five) designed to assist in understanding what they think about the data collection day. Measures of frequency (percentage) and central tendency (mean) were adopted for descriptive analysis. Six of seven contacted schools accepted participation in the project. All parents who attended the explanatory meetings agreed to join the project. A cohort of 676 students (336 girls) participated at baseline enrollment, with a follow-up rate of 96% in the second data collection. The average questionnaire-related scores were 4.2 ± 0.7 and 4.4 ± 0.6 for teachers and students, respectively. These 3D craniofacial norms will benefit multidisciplinary teams managing cleft-craniofacial deformities in the globally distributed ethnic Chinese population, particularly useful for phenotypic variation characterization, conducting quantitative morphologic comparisons, and therapeutic planning and outcome assessment. The described pathway model will assist other groups to establish their own age-, sex-, and ethnic-specific normative databases.

## 1. Introduction

Patients with a myriad of congenital cleft-craniofacial deformities or acquired abnormalities routinely undergo surgical treatment from infancy to maturity with the primary goal of restoring functionality as well as craniofacial form, proportion, and symmetry [1,2,3]. Seminal anthropometric and two-dimensional (2D) photographic studies documenting craniofacial norms have been essential for cleft-craniofacial practice as well as for research purposes [4,5,6,7]. However, these traditional methods are not ideal for comprehensively appraising the craniofacial region, which is a complex three-dimensional (3D) structure [8,9,10].

In this setting, non-contact 3D surface capturing system technology has several advantages over the former direct and indirect anthropometric techniques, including quick capture speed of high-quality images without distortion of spatial form, shape, and size, resulting in a lifelike rendering; the ability to extract x, y, and z coordinate data; the production of an excellent photorealistic bitmap to assist with landmarking and segmentation; the lack of any ionizing radiation; and a high degree of reproducibility, reliability, and validity [8,9,10]. Therefore, a growing number of 3D craniofacial soft-tissue norms have been established for age-, sex-, and ethnic-specific populations in an effort to overcome many of the limitations in the traditional normative datasets [11,12,13,14,15,16,17,18,19].

Most previous 3D craniofacial surface imaging norms or average children’s faces have predominantly been based on descriptions of cross-sectional study designs or mixed longitudinal and cross-sectional designs [11,12,13,14,15,16,17,18,19]. Longitudinal studies are often considered the gold standard for characterization and quantification of craniofacial morphological development from infancy to maturity, providing vital information for understanding the progress of population- and patient-specific issues with establishment of patterns of growth and directional and fluctuating asymmetry. However, a long-term longitudinal study requires measurements of the same individuals on multiple occasions over a period of years, which is associated not only with barriers to recruitment but also with the risk of bias due to incomplete follow-up [20,21,22]. Different strategies have been described in the literature to overcome these recruitment- and retention-related issues, including flexibility in data collection methods, offering assistance with transport, novel methods of engaging participants (e.g., web advertising, social media, and electronic reminders), sharing study results, and reminder strategies (e.g., phone calls, SMS (short message service), and email) [23,24,25]. Particularly, the operational characteristic of 3D craniofacial surface imaging-based longitudinal studies has not yet been formally addressed in detail.

The purpose of this study was to describe the workflow and strategies used to enhance recruitment and retention of healthy Taiwanese Chinese elementary school students in the first three years of a longitudinal study involving annual 3D craniofacial imaging. The implementation of this sex-, age-, and ethnic-specific 3D normative craniofacial soft-tissue dataset from healthy population-representative samples has broad application in research and clinical practice encompassing the globally distributed ethnic Chinese population, such as supporting syndrome delineation and early diagnosis of congenital and genetic disorders that modify facial phenotype, for early detection of craniofacial soft-tissue disturbances related to development, for quantitative morphologic comparisons, and in therapeutic planning and outcome assessment. These normative data would act as key drivers for scientific advances based on multidisciplinary collaboration between computer sciences, bioengineering, epidemiology, genetics, dentistry, orthodontics, and surgical disciplines.

## 2. Materials and Methods

This study was performed in the Imaging Laboratory, Craniofacial Research Center, Chang Gung Memorial Hospital, Taoyuan, Taiwan. This is a methodological description of a longitudinal study for quantification of the craniofacial morphologic development. This project intended to acquire annual 3D images from elementary school students aged 6 to 12 years at baseline until they reached skeletal maturity (Figure 1). The normative World Health Organization standard values for Taiwanese Chinese population was adopted for definition of skeletal maturity [26].

### 2.1. Inclusion and Exclusion Criteria

Healthy pediatric individuals, aged between 6 and 12 years, were recruited on a voluntary basis from elementary schools. Potential schools were randomly selected among the existing private and public elementary schools neighboring the Linkou or Taoyuan branches of the Chang Gung Memorial Hospital, Taoyuan, Taiwan.

Each potential participant was clinically screened extra- and intra-orally by craniofacial orthodontic and craniofacial plastic surgeon professionals. Participants were excluded from the study if they had (1) mixed or uncertain ethnicity (non-Taiwanese Chinese); (2) presumed or confirmed diagnosis of any syndromic or non-syndromic cleft-craniofacial deformity; (3) a history of cranial or facial trauma; (4) underwent orthodontic treatment or any cranial, facial, or oral surgical intervention; or (5) occurrence of cleft-craniofacial deformity in first, second, or third-degree relatives. No socio-economic or school grade-related status parameters were included as selection criteria in this study.

### 2.2. Project Workflow

From the initial year of project definition to the first two-year data collection, multi-faceted approaches were implemented and adjusted to ethically and safely recruit the students for baseline data collection and to maintain their participation in subsequent years. The key aspects for project delineation, implementation, and annual data collection from 2016 to 2018 are depicted in an integrated multistep workflow in Figure 2.

### 2.3. Ethics

The research project was reviewed and approved by the Human Research Ethics Committee (Chang Gung Memorial Hospital, protocol: 201601192B0) with consideration of risks, discomforts, and benefits in confluence with the study’s objective. The assent and consent forms provided details about the project’s interviews, examinations, measurements, and storage of 3D craniofacial image material. Introduction of the study, objectives, institution involved, and participants’ rights (including stopping participating in the research at any moment) were also addressed. This ethical process provides confirmation of safety (noninvasive measures for image and information acquisition and insurance coverage), benefits (educational measures), confidentiality, secrecy, subjects’ protection during data collection, storage, and processing (without personal identifiers such as name, age, address, and status of the oral healthy, occlusion or psychosocial elements), and dissemination of the results (deidentification of data) [27]. Only authorized personnel could access the encrypted system’s database.

### 2.4. Budgeting Strategy

The main funding was provided for the full long-term project proposal and further specific grant extensions for each three-year period were possible. Overall, the funding was stipulated for covering staff salary (research assistants), equipment, supplies, and services acquisition (e.g., 3D imaging-, office-, and audiovisual-related materials, toothpaste, and toothbrushes, among others), and cost of parents/students’ participation (e.g., insurance coverage, transportation, and lunchbox). No monetary incentive was offered for participation in the study at baseline or subsequent enrollments.

### 2.5. Engagement of Partners and Students

The deans of the elementary schools and administrator staff acknowledged the project by initial contact via e-mail and telephone calls, and accepted the project after a descriptive meeting with our research team. We then conducted explanatory meetings with the pre-defined partners of research, i.e., school stakeholders (dean, administrator staff, and teachers) and parents, to explain the study purpose, the intervention content, and the direct potential benefits of students’ participation, as well as the benefits of the new knowledge from the research. These meetings also functioned to answer questions, address concerns, and establish strategies to minimize potential impact on regular school functioning. As a key part of this engagement activity, our pre-defined recruitment and retention strategies were discussed with the partners of the project (school stakeholders and parents) who helped brainstorm solutions to challenges that would be faced before and during data collection progression. We considered students’ participants needs and preferences, including specific educational measures (i.e., concepts such as oral hygiene, psychosocial interaction among peers, and the necessity to increase awareness about cleft-craniofacial conditions among members of the public), and the data collection period separate from the regular school activities. All partners had access to the research team office’s contact phone number and e-mail address, providing opportunity for frequent contact that would facilitate the process. An instant messaging app (LINE Messenger, LINE Corporation, Tokyo, Japan)-based chat group (named the Normative 3D Craniofacial Project in Taiwan) was specifically created to engage in synchronous text-based conversations among those involved in this project.

Subsequently, a didactic approach with audio-visual material was adopted to contact the students. All topics presented, discussed, and defined with the partners were formally explained and interactively reviewed with the students. Their opinions and concerns were considered, and the updated project design was then confirmed. The engagement of partners and students with the understanding of project issues created a virtuous cycle of commitment involved participation and retention in the study.

### 2.6. Research Staff Members

The professional staff included craniofacial orthodontists, pediatric dentists, craniofacial plastic surgeons, pediatric physiotherapists, a bioengineer, and informatic-trained specialists. Laboratory technicians and medical and dental students were employed as complementary assistants. A specific researcher assistant was defined as the recruitment coordinator who functioned as a central team member for monitoring progress and maintaining open lines of communication with every partner or student. Before each data collection day, we revised the overall step-by-step process and the particular task for each professional for quality assurance and to identify challenges and solutions based on staff members’ experiences in their own practice and research activities.

### 2.7. Tripod Station-Based Data Collection Day

The data collection day was organized with three different stations, including two educational stations and a 3D imaging station. All participants were assigned to start at one of three stations using identification badges (Figure 3). As an activity was completed, each group of students was moved to the next station until all had completed the three activity stages. Each station lasted about 1–1.5 h, with the three activities delivered in the morning with lunchbox offered at the end of the activities. For this, all students, parents, and teachers (if they wanted to participate in the actions) were transported by a specialized student transportation company from the school to the hospital (early in the morning) and then returned to school after activities.

Different data collection days were dispersed throughout the year according to the requests and requirements of school stakeholders, parents, and students, ensuring no interference with the regular school activity calendar. Additional flexibility in rescheduling data collection times was then provided for participants who missed one specific data collection day.

### 2.8. Oral Hygiene Educational Station

At this station, an initial educational lecture covered the importance of oral health and prevention methods, using illustrative and interactive slideshows and videos as support material. Dental health behavior, including use of tooth paste, tooth-brushing time, and method, were formally addressed [28,29,30]. Further discussion concerning students’ attitude and feelings regarding dental health was simulated using a brainstorming approach. Dental model-based tooth-brushing demonstration was an additional applied educative method. The participant then brushed their teeth under the supervision of a dental practitioner, who corrected any mistakes and re-trained the right movements, if needed. All students had their oral hygiene and occlusal status diagnosed by pediatric dentistry, with referrals performed according to the parents’ acceptance and individual needs, e.g., low oral hygiene levels, inadequate brushing skills, and dental caries.

### 2.9. Psychosocial Educational Station

The activities in the psychosocial interaction educational station were divided in two parts: six-piece story-making and stigmatizing and discrimination issues of cleft-craniofacial deformities. Pediatric psychotherapists initially encouraged all students to participate in six-piece story-making [31,32,33] to let them explore themselves. Each student was asked to (1) create a main character/protagonist (not necessarily human) in a fictional, fantasy, or historical setting. The instructions lead the student on through (2) the creation of a task for the main character, (3) obstacles they encounter, (4) helpful factors, (5) the climax of the story, and (6) its conclusion (Figure 4 and Figure 5). The interpretation of this six-piece story allows a multi-level understanding of the student. The assumption is that the themes, conflicts, world view, and problem solving that are displayed in the story communicate something meaningful about the student’s own experience. Erikson’s stages of psychosocial development [34], i.e., adolescence (6 to 12 years; adopted at baseline) and teenage (13 to 20 years; adopted in follow up data collection), were employed as basis for interpretation and discussion of this process. In each stage, the child confronts new psychosocial challenges/crises and either masters or fails to master them. The pediatric physiotherapists appraised the process to draw a map of the student’s inner world. The Erikson’s classification tool adopted in this station also allowed for screening of a wide spectrum of psychosocial problem such as the autism spectrum disorder. The option to be referred to additional psychodynamic diagnostic evaluation was formally provided to parents of students who did not have coping resources with age-oriented conflicts and questions. All parents requesting further information also received personalized description about their child.

Elementary school is a crucial period during socialization, and often students with head and/or facial differences are labeled as abnormal and could suffer from bullying from their peers. Therefore, the other particular element of this educational station was aimed to raise awareness related to stigmatizing and discrimination in healthy students by providing simple but adequate information about relevant issues (social stigmatization, discrimination, misperceptions, misconception, and misinformation) surrounding cleft-craniofacial deformities. Colloquial words, i.e., exclusion, rejection, avoidance, isolation, facial difference, mark on the face, facial scar, among others, were employed to facilitate communication among research staff and participants in all activities that were delivered in Mandarin. This educative initiative was part of a multi-level stigma intervention [35,36]; therefore, there was no intention to fully address the stigmatizing attitudes and behaviors within the whole community. This had the potential to raise awareness among students and importantly to introduce discussion about this topic at home and school environment between stakeholders, parents, and students.

### 2.10. 3D Data Acquisition

3D craniofacial stereophotogrammetric surface images were acquired for each participant using the 3dMD system (3dMD LLC, Atlanta, GA, USA) under the following standard conditions: a permanent installation with fixed ambient lighting and system and fixed individual positioning, including individuals with a natural head position, relaxed facial musculature, a closed mouth, and thin elastic nylon caps to keep the hair away from the face [2,10,13]. The system was calibrated before every capture process. The image set was immediately evaluated by a team member and repeated as necessary to obtain an acceptable image (Figure 6). The 3D image dataset was stored in the Imaging Laboratory of Chang Gung Craniofacial Research Center (Taoyuan, Taiwan). This raw 3D image dataset can be constantly accessed and tailored to fit future study designs, including morphometric quantification of different regions of interest (e.g., full craniofacial, face, and nasolabial regions) as well as characterization of fluctuating (from the less to most asymmetric pattern of presentation) and directional (right- or left-sided predominance of prominent cranial and facial contour) asymmetry parameters within the healthy sample.

### 2.11. Reinforcement Tactics

Reinforcement tactics were applied to sustain the annual participation after the first enrollment. Constant communication was maintained between and within school stakeholders, parents, and students regarding comments and criticisms related to each particular experience, with adjustments performed accordingly. Our research team sent reminder letters, emails, and app-based messages 6 months, 3 month, 1 month, and 1 week before the second data collection day. Participants who did not attend the pre-scheduled second data collection day were further contacted by school stakeholders via phone calls and text messages. All parents were asked to provide contact information, i.e., postal and e-mail addresses, phone number, and LINE ID (LINE Messenger, LINE Corporation, Tokyo, Japan), during the baseline enrolment, with all data being updated during the second data collection. The study progress was shared with participants via feedback reports using a messaging app-based chat group and customized visual material. Within one month after the data collection day, each participant received DVD material displaying their own 3D craniofacial images. Only this material had a 3D image exhibiting the actual face of students.

### 2.12. Participation-Related Satisfaction Questionnaire

Students were randomly selected among participants regardless of the school of origin to complete an eight-question survey with five-point Likert scales (range from 1 to 5) regarding their experiences during the data collection day. Randomly selected teachers from all of the participant schools were also requested to participate of this survey (Appendix A). Items were carefully worded for comprehension by the target cohort, with further clarification provided by the study staff. Only age and sex information were provided with no name or school identification.

### 2.13. Statistical Analysis

For the descriptive analysis, the mean was used for metric variables (age and satisfaction questionnaire), and percentages were given for categorical variables (gender and number of participants).

## 3. Results

### 3.1. Partners of Project

During the first year of this longitudinal study, in addition to the preparation of structural and professional staff elements, we approached potential partners for the project. Six of seven elementary schools contacted by e-mail and telephone call provided agreement to receive our research team for delivering additional information. All six schools that attended our descriptive meeting accepted participation in the project, allowing and aiding us to initially contact teachers and parents. All parents and students who attended our explanatory meetings accepted participation in the project as ethically provided through signature of informed consent and assent forms.

### 3.2. Study Subjects

For the initial acquisition of information, nine days were specifically delineated for data collection, which were distributed over time to meet the requests and requisites of partners and students. A cohort of 676 elementary school students (aged 6 to 12 years; 336 girls) participated in this baseline data collection. The follow-up rate was 96% for the second annual data collection. All students participated of the three stations of data collection day. Participants (4%) who did not attend in the second data collection day have changed to schools that were not part of the initial school supporters of the project. No request for withdrawal was received in this initial two-year period.

### 3.3. Satisfaction Questionnaire

Eight teachers (mean age of 34.4 ± 7.3 years; 63% women) from each included elementary school and 47 students (mean age of 9.6 ± 0.5 years; 55% boys) from different elementary schools answered the participation-related satisfaction questionnaire. The overall average scores were 4.4 ± 0.6 and 4.2 ± 0.7 for teachers and students, respectively (Table 1).

## 4. Discussion

The current longitudinal study design-based project targeted the establishment of the first 3D image-based craniofacial soft-tissue norms database in Taiwanese Chinese elementary school students. For this, after active engagement of partners and the initial recruitment of healthy population-representative samples, a high retention rate was achieved during the second round of data collection. As noted by other longitudinal studies [23,24,25,37,38,39], recruitment and retention efforts were successful due to comprehensive planning with a multifaceted approach. Our multimodal strategies were designed based on the historical challenges faced in conducting longitudinal research [23,24,25,37,38,39], with amendments applied as the project proceeded. We stayed in consistent contact with the partners of the project and students concurrently with ongoing different data collection days, which permitted the interactive diagnosis of potential or actual issues and the implementation of solutions accordingly. We mainly perceived issues during the first (initial planning and contact with partners of research and students) and second (baseline enrollment) years of study, with a more efficacious workflow during the third (second enrollment) year.

Overall, we aimed to provide an interesting, educative, friendly, accessible, and recognizable project. Several of adopted strategies were validated in previous studies [23,24,25,37,38,39], including engagement of participants, branding tactic (name of project in messaging app-based group chat), short distance to travel with project-supported transportation, activities performed with flexible scheduling, continuous contact with the target population, aided participant convenience with the use of a dedicated contact line, and delivery of other activities in addition to the data collection. Personalized communication strategies, i.e., specific chat group and DVD-based visual material, also encouraged the interest of partners and students about the project progress as well as for setting the study apart from unsolicited broad-based communications like spam and telemarketers.

Partner engagement actions that recognized students’ barriers and facilitators for participation were key elements to achieving baseline enrollment as well as the high retention rate. Besides the described strategies, our research team noticed that the parents created their own messaging app-based chat group for exchange of project-related information. The multi-language LINE platform (LINE Messenger, LINE Corporation, Tokyo, Japan) is the one of most popular free messaging apps for a variety of smartphones and computers in Asia. Due to its several features and applications, e.g., media, communication, social networking, and e-commerce, this app has become a practical tool in daily Taiwanese lives and activities, which positively contributed to the dissemination of project-related information. The broadcasting of pre-defined data collection days and possibility of rescheduling as needed helped overcome some of obstacles by providing parents with the flexibility and convenience to create their own calendar for scheduling their children’s participation. Differences among Western and Eastern cultures, i.e., dimensions of individualism (individual initiative and independence) versus collectivism (group solidarity and collective identity) [40] when using messaging apps should be considered when appraising the participation pattern described in this project.

Another factor that may have positively influenced parents’ engagement and students’ participation is the study topic, which was the evolution of 3D craniofacial images over time. The face is a relevant issue into Asian society as it is of interest not only to researchers, sociologists, and health-care professionals, but also broadly to public [41,42]. In Asia, facial physiognomy and facial features are important in daily life, as self-confidence and marriage and career prospects are influenced by appearance [41]. Assessments of individuals’ mental or moral character, fortune, and future are often judged based on facial features [41,42]. Surgical-derived facial changes are considered to positively impact personal and professional opportunities and relationships [42,43]. Therefore, facial surgical treatments are popular throughout Asia, especially during younger ages [43]. Students who have finished the college or university often receive facial surgical interventions before they actually start their relationships in a new social environment. The degree that this sociocultural behavior influenced parents’ and students’ decisions to participance in this craniofacial image-centered project should be a subject of further investigation.

The use of high-technology-based facial imaging in this project may also be considered an additional component influencing student desire to participate. Currently, students are highly connected through the Internet, tablets, and mobile phones with wide use of social networking sites [44,45]. Therefore, they may have been curious, attracted, and motivated about the possibility to be part of a project targeting advanced technology with computer-assisted 3D craniofacial image generation. The visualization of their own face in high-quality, full-colored, and realistic 3D images with the possibility for interactive rotation of the face in many directions after the first enrolment may have further positively influenced students’ commitment to attend the second data collection.

This descriptive study is not without limitations. The absence of descriptions related to strategies of recruitment or retention rates in previous 3D normative database impair a reliable head-to-head comparison between the existing studies and our current results. We do not have information about the number or reasons of potential parents or participants who did not join to these meetings, as we were committed along with the school stakeholders to truly respect the refusals and decisions of parents/students, and we did not approach for formal ethical-supported approval. All parents who attended to the explanatory meeting provided consent for their children be participate in the project, but we did not collect information about the reasons for participation. Despite this lack of reason-related data, the school stakeholders described that positive feedback was provided by the teachers, parents, and students after the data collection day experiences. The questionnaire-related results corroborate this information as, on average, the randomly selected teachers and participants were satisfied with the experience and reported that they want to participate again in future years. We did not administer the questionnaires to all participants to reduce the burden of the many activities in research projects and to not increase the overall time commitment required during the data collection day. Future alternative surveys, i.e., Internet-based questionnaires, could maintain these requirements an increase of representativeness of the total sample.

An additional limitation is the lack of specific controlled information about the distinct effect of individual strategies on enrollment and retention; because we adopted a diverse range of strategies and allowed them to evolve over time, we cannot draw any conclusions about the actual impact of any single strategy. This study has a descriptive methodological characteristic and some judgements about the importance of, for example, partners’ engagement through the messaging app, were based on reflections from the research team during the research progress, which deserve further deep investigation. The described workflow should not be interpreted as unique or absolute and additional improvements and amendments are part of our constant initiatives to improve participation and meet the needs of students over time. As the missing data in the second round of data collection were due to students who moved to schools that were not part of project, we needed to amend our project to minimize this issue in the coming years.

This was a longitudinal project to monitor craniofacial form over time; therefore, achieving retention success requires continually monitoring progress at every level of a project with implementation of adapted strategies accordingly, including other age-specific educational measures as the elementary school students age. Other groups using 3D imaging technology and planning to set normative ethnic-specific datasets could adopt the described tactics as a base to continue evolving and refining longitudinal data collection according to the particular needs and expectations of their population and environment. This could culminate in the creation of a large multi-ethnic database in the future.

## 5. Conclusions

This study described the workflow for the establishment of 3D craniofacial soft-tissues norms in Taiwanese Chinese individuals. The portrayed strategies can assist other groups to start planning data collection for establishment of their own age-, sex-, and ethnic-specific normative database.

## Figures and Tables

**Figure 1 ijerph-16-04438-f001:**
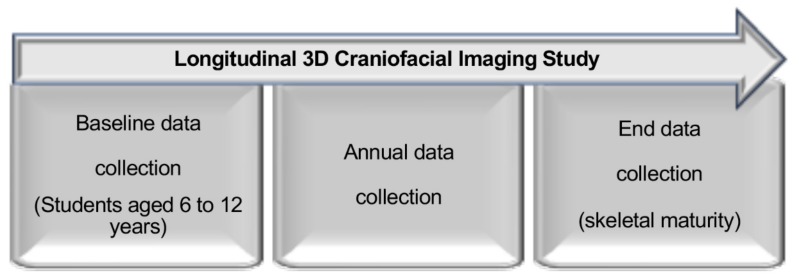
Flowchart of the data collection timeline.

**Figure 2 ijerph-16-04438-f002:**
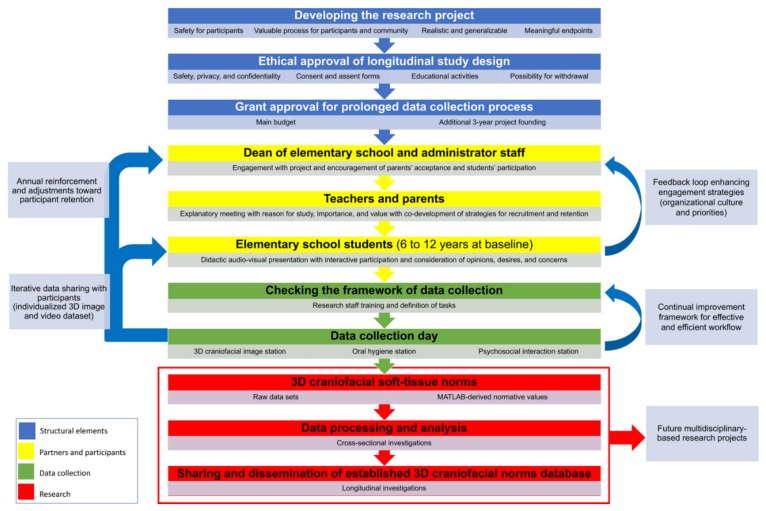
Workflow and strategies adopted for recruitment and retention in this longitudinal three-dimensional (3D) craniofacial imaging study.

**Figure 3 ijerph-16-04438-f003:**
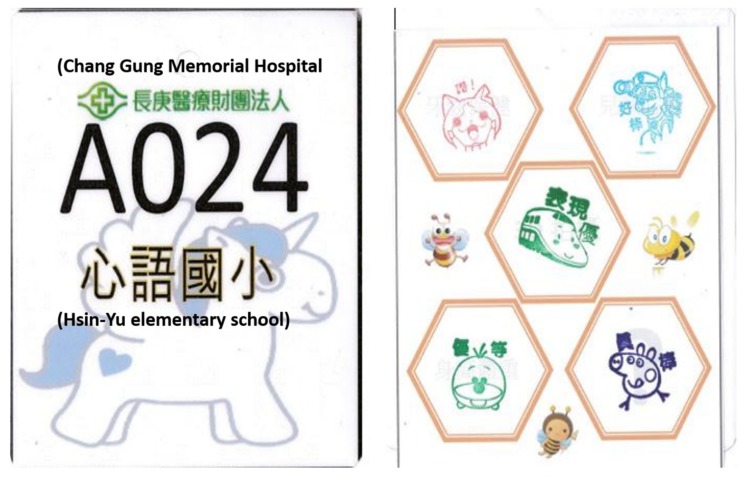
Practical example of the identification badge adopted to assist in the flow process between the stations on the data collection days. Translation from Mandarin Chinese to English: Chang Gung Memorial Hospital (green color) and Hsin-Yu elementary school (black color).

**Figure 4 ijerph-16-04438-f004:**
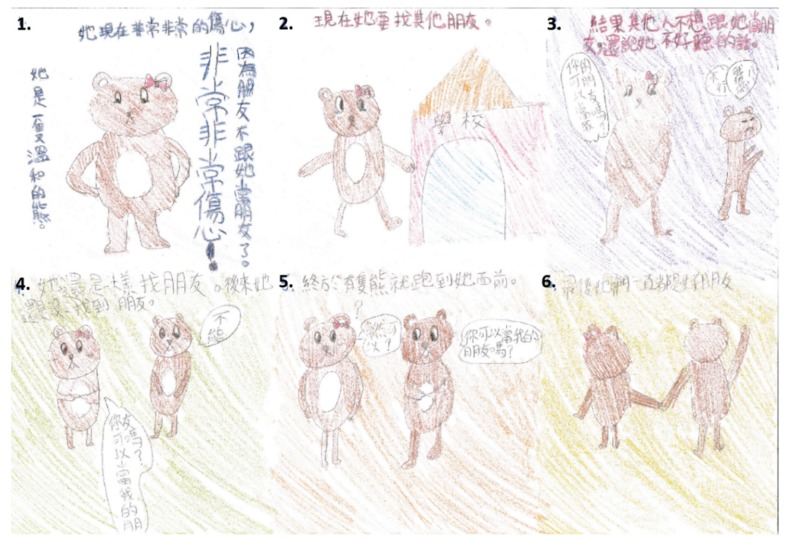
Practical example of six-piece story-making. Translation from Chinese Mandarin to English: (1) She was a very calm bear. She was very sad then, because no one wanted to make friends with her. (2) In the school, she hoped to find some other friends. (3) However, they did not want to make friends with her and said bad words to her. She said: can I make friends with you? The other bear replied: no way, I hate you. (4) She tried to look for other friends, but in vain. She said: can I make friend with you? The other bear replied: no, I can’t. (5) Finally, one bear came to her front to say: can you be my friend? She replied: of course. (6) To date, they are always good friends.

**Figure 5 ijerph-16-04438-f005:**
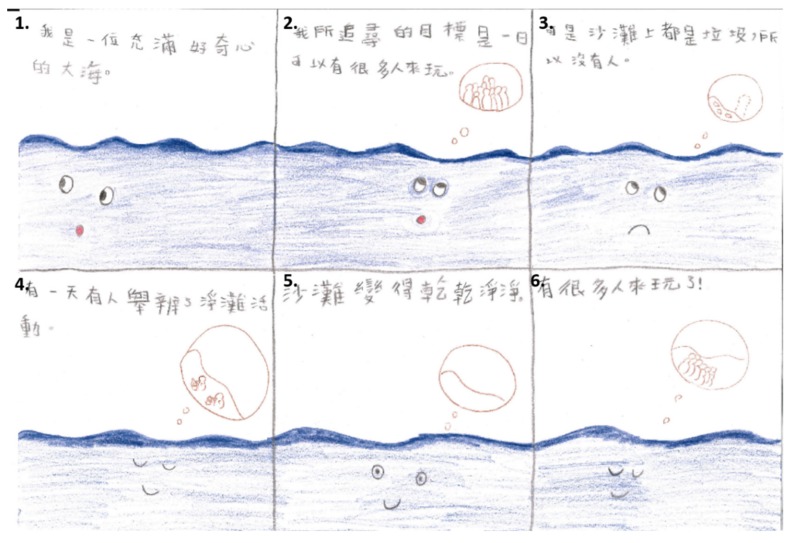
Practical example of six-piece story-making. Translation from Chinese Mandarin to English: (1) I was a fully curious ocean. (2) My goal was that someday everyone can come play. (3) But there was no one here due to garbage all around. (4) One day, someone held a clean-beach activity. (5) The beach became very clean. (6) A lot of people came play.

**Figure 6 ijerph-16-04438-f006:**
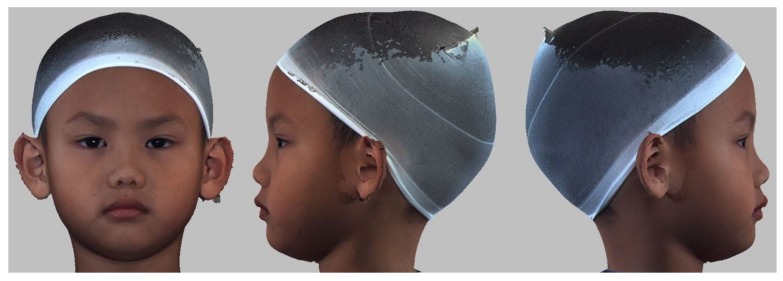
Three-dimensional (3D) craniofacial image from an elementary school student.

**Table 1 ijerph-16-04438-t001:** Results for participation-related satisfaction questionnaire among teachers and students.

Question *	Teachers (mean ± SD)	Students (mean ± SD)
1	4.4 ± 0.5	4.0 ± 0.7
2	4.4 ± 0.7	4.1 ± 0.7
3	4.5 ± 0.5	4.3 ± 0.6
4	4.3 ± 0.7	4.3 ± 0.6
5	4.8 ± 0.4	4.2 ± 0.7
6	4.6 ± 0.5	4.1 ± 0.7
7	4.4 ± 0.5	4.3 ± 0.7
8	4.1 ± 0.8	4.3 ± 0.6

* Five-point Likert scale ranging from 1 to 5.

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
