# Peer review of "Workflow and Strategies for Recruitment and Retention in Longitudinal 3D Craniofacial Imaging Study"

_ijerph, 2019, doi:10.3390/ijerph16224438_

Round 1
Reviewer 1 Report
Thank you for submitting your work.
The authors trying to share their experience in collecting 3D images of children for longitudinal study, which worth to be published as it can be of benefit to other readers.
I have the following comments to improve the quality of this work.
The title is too long, it has to be shorter, and specific. Avoid the use the word ''normal'' in describing the healthy baby, for example in line 96, please delete the word normal and replace it with healthy. line 305, the authors gave the scores of the questionnaire as 4.4 but they did not mention whether this score was out of 5 or out of 10, the same problem is in table 1. Please, mention whether this score is out of 5 or out of 10 as this would make a big difference. The questionnaire is not available in the manuscript, would you please add it as an appendix. English editing is necessary, the authors tried to use very long sentences with are confusing, for example, line 89:''This is a methodological description of a longitudinal study design involving the acquisition of annual 3D images for quantification of the craniofacial morphologic development from elementary school students aged 6 to 12 years at baseline until they reach skeletal maturity, as defined in the normative World Health Organization standard values for Taiwanese Chinese population [25].''
Reviewer 2 Report
A WORK FLOW DIAGRAM OF DATA COLLECTION TIME LINE WILL BE HELPFUL TO BE ADDED SUCH AS
TIME 1 : BASE LINE 1
TIME 2 : BASE LINE 2
TIME 3 : ONE YEAR FOLLOW UP
THIS WAY IT WILL HELP IN ILLUSTRATING THE PROJECT
DESCRIBE MORE ON HOW ERIKSON CLASSIFICATION WILL HELP IN NOT ONLY ASSESSING PSYCHOSOCIAL BUT ALSO ESTABLISHING SCREENING FOR PSYCHOSOCIAL PROBLEM IF ANY
SUCH AS POSSIBLE AUTISM SPECTRUM, OR ANY OTHER PSYCHOSOCIAL DISTURBANCES
Reviewer 3 Report
Interesting report,
In the abstract, you need to provide more on the statistical tests and the findings
Introduction, you need to include soft tissue longitudinal studies as well not just reference 4-6, add studies such as Angle Orthod. 2014 Jan;84(1):48-55.
you need to work methodology and expand it more
Figure 1 needs some magnification, very difficult to read
Section 2.3. Ethics should be at the start of M&M section
Section 2.4. Budgeting Strategy can be added to the acknowledgement and funding section at the end
you need a section(sub-heading) on 'statistical analysis'
what was the 'the participation-related satisfaction questionnaire. " about , this is not clear for the reader, need more on 'five-point Likert scales ' 1-5 , or 0-4?
where are the findings on '3D Data Acquisition' is this gonna be reported later on?
section sub-heading '3.2. Students' is confusing, change to 'study subjects'
Round 2
Reviewer 3 Report
thank you for the revisions,
abstract
as a reader, I struggle to see the main findings of the study, what is the take-home message, the conclusion is very general, rewrite, what are the strategies that you found beneficial in your cohort ? add female /male numbers for the cohort
very deficient literature review, line 72, Different strategies have been described in 72 the literature to overcome these recruitment- and retention-related issues [23–25].' mentions these strategies for reader
what do you mean by'Randomly selected students and teachers answered an experience satisfaction questionnaire using a five-point Likert scale.' satisfaction about what? explain to reader
figure 1 is too big and needs revision
figure 2 is too small and difficult to read, modify
add a section showing the 'the participation-related satisfaction questionnaire'
2.13. Statistical Analysis, expand this section what variables you intended to record, measure?
discussion is not good, you need to talk about the findings of this study and compare with other similar studies, talk about difficulties, and again what are these strategies
